# Hybrid Complex Coacervate

**DOI:** 10.3390/polym12020320

**Published:** 2020-02-04

**Authors:** Marco Dompé, Francisco Javier Cedano-Serrano, Mehdi Vahdati, Dominique Hourdet, Jasper van der Gucht, Marleen Kamperman, Thomas E. Kodger

**Affiliations:** 1Laboratory of Physical Chemistry and Soft Matter, Wageningen University & Research, 6708 WE Wageningen, The Netherlands; marco.dompe@wur.nl (M.D.); jasper.vandergucht@wur.nl (J.v.d.G.); marleen.kamperman@rug.nl (M.K.); 2Soft Matter Sciences and Engineering, ESPCI Paris, PSL University, Sorbonne University, CNRS, F-75005 Paris, France; francisco.cedano@espci.fr (F.J.C.-S.); mehdi.vahdati@espci.fr (M.V.); dominique.hourdet@espci.fr (D.H.); 3Laboratory of Polymer Science, Zernike Institute for Advanced Materials, University of Groningen, 9747 AG Groningen, The Netherlands

**Keywords:** complex coacervation, nanofillers, nanocomposites, polyelectrolytes, underwater adhesion, poly(*N*-isopropylacrylamide)

## Abstract

Underwater adhesion represents a huge technological challenge as the presence of water compromises the performance of most commercially available adhesives. Inspired by natural organisms, we have designed an adhesive based on complex coacervation, a liquid–liquid phase separation phenomenon. A complex coacervate adhesive is formed by mixing oppositely charged polyelectrolytes bearing pendant thermoresponsive poly(*N*-isopropylacrylamide) (PNIPAM) chains. The material fully sets underwater due to a change in the environmental conditions, namely temperature and ionic strength. In this work, we incorporate silica nanoparticles forming a hybrid complex coacervate and investigate the resulting mechanical properties. An enhancement of the mechanical properties is observed below the PNIPAM lower critical solution temperature (LCST): this is due to the formation of PNIPAM–silica junctions, which, after setting, contribute to a moderate increase in the moduli and in the adhesive properties only when applying an ionic strength gradient. By contrast, when raising the temperature above the LCST, the mechanical properties are dominated by the association of PNIPAM chains and the nanofiller incorporation leads to an increased heterogeneity with the formation of fracture planes at the interface between areas of different concentrations of nanoparticles, promoting earlier failure of the network—an unexpected and noteworthy consequence of this hybrid system.

## 1. Introduction

Adhesive technology, despite the tremendous progresses obtained in the automotive and construction industrial fields in the last century [1], is rarely applied in wet environments: the presence of water is detrimental to the performance of commercially available adhesives, promoting swelling of the material or weakening the contact with the adherend [2]. For this reason, the adhesive application in challenging environments, such as the human body, is strongly limited [3]. Yet, the replacement of mechanical fasteners (e.g., sutures and staples), being beneficial both for the patient and the surgeon, would lead to an easier application, a shortening of the operating times and a lower chance of infection [4,5]. Commercial products, such as cyanoacrylates [6,7], fibrin glues [8,9], and PEG-based adhesives [10,11], fail to meet the requirements needed for tissue adhesives [5,12]—mainly easy delivery, fast setting times, high cohesive and adhesive properties in wet environments and lack of cytotoxicity—and, therefore, new design principles are needed.

Natural seawater organisms, striving to survive in harsh conditions, have solved these issues [13]. Mussels strongly attach to several surfaces using a network of threads, enabling them to resist the huge forces exerted by waves [14]. Sandcastle worms build protective tubules collecting sand grains and connecting them with a proteinaceous glue, which hardens underwater due to a change in the environmental conditions (change in pH and ionic strength, exposure to oxygen) [15]. Complex coacervation, an associative liquid–liquid phase separation mainly driven by electrostatic interactions [16,17], is believed to play a crucial role in the processing of the natural glues: the sandcastle worm, for example, stores oppositely charged polypeptides in granules, which rupture upon delivery in seawater, releasing the adhesive which hardens over time due to a combination of physical and covalent bonds [18,19]. Complex coacervates are concentrated phases of oppositely charged polyelectrolytes which are promising candidates for underwater adhesion as they are immiscible with water [16], possess low interfacial tension enabling good wetting of the surface [20] and have high dimensional stability [21]. Additional interactions need then to be introduced to turn the material into a tough solid which can oppose detachment. Several researchers have managed to provide the required properties by using strong oxidizing agents [22,23,24]: however, these are potentially toxic and lead to chemical crosslinking of the material, with the loss of viscoelasticity, a feature that enables energy dissipation. An alternative strategy toward toughening could be the incorporation of hard nanoparticular fillers which do not possess these toxic side-effects.

We have recently reported the development of a complex coacervate-based adhesive which sets underwater only because of a change in environmental conditions, namely temperature and ionic strength, exclusively forming physical bonds [25,26]. The material is obtained by mixing oppositely charged polyelectrolytes modified with pendant poly(*N*-isopropylacrylamide) (PNIPAM) chains: PNIPAM is a well-known thermoresponsive polymer, which undergoes phase separation when surpassing its lower critical solution temperature (LCST), which is generally approximately 32 °C. [27] When tested in physiological conditions, the material is held together by different types of non-covalent interactions, giving rise to variable bond strengths, with strong electrostatic interactions between the polyelectrolyte backbones imparting elasticity and PNIPAM bonds breaking and dissipating energy. This leads to an increase in toughness [28] and to promising underwater adhesive performance: however, to fully function as a tissue glue, the adhesive strength needs to be further improved.

In this work, we aim to enhance the mechanical properties by homogeneously incorporating a nanofiller within the complex coacervate, leading to the development of nano-reinforced hybrid adhesives [29]. Nanoparticles, with dimensions smaller than 100 nm, such as nanoclays, nanosilica, carbon-based and metal-based nanoparticles [30,31,32,33], are generally used as reinforcing agents in rubbers for example: compared to bulk materials, nanoparticles have higher specific surface area and surface energy, together with a lower amount of structural imperfections [29]. The introduction of nanofillers in the adhesive matrix may enhance the material stiffness mainly due to a restricted mobility of the polymer chains: nanoparticles, generally with a higher stiffness than the polymer matrix, end up in the spaces between the polymer chains, reducing their flexibility [29]. Many researchers have also reported an improvement in fracture toughness, which is ascribed to the introduction of strain-dependent damage mechanisms, which enable energy dissipation [34].

The extent of dissipation depends on many factors, which mainly are the nature of the filler (type, shape, size, charge density), its volume fraction and its interaction with the matrix: a basic requirement that always needs to be fulfilled is a homogeneous dispersion of the filler within the material [34]. However, this is challenging as, due to the small size and large surface area, nanoparticles are subjected to strong attractive forces, leading to the undesired formation of aggregates which undermine the mechanical properties [29]. In order to circumvent this issue, we take advantage of the specific interactions existing between PNIPAM and silica nanoparticles [35,36]: poly(*N*-alkylacrylamides) are known to form protective layers around particles, useful for steric stabilization, and to strongly bind to inorganic surfaces [37,38]. These physical crosslinks between inorganic beads and polymeric matrix have led to the development of nanocomposite hydrogels with exceptional mechanical properties and of nanoparticle-based adhesives for biological tissues [39,40,41,42]. In this work, we employ these PNIPAM–silica interactions to promote the nanoparticle incorporation within the complex coacervate matrix and successively evaluate the enhancement of the rheological and adhesive properties.

## 2. Materials and Methods

### 2.1. Materials

Poly(acrylic acid) (PAA, analytical standard, *M_n_* = 239 kg/mol, *M_w_* = 1030 kg/mol), poly(*N*-isopropylacrylamide) amine terminated (PNIPAM–NH_2_, average *M_n_* = 5.5 kg/mol), *N*,*N*’-dicyclohexylcarbodiimide (DCC, 99%), acrylic acid (AA, 99%), potassium persulfate (KPS, ≥99%), *N*-methyl-2-pyrrolidone (NMP, anhydrous, 99.5%), sodium chloride (NaCl, ≥99%), LUDOX^®^ TM-40 colloidal silica (40% wt. suspension in H_2_O, particle radius = 9.8–13.4 nm, specific surface area = 110–150 m^2^/g, pH = 8.5–9.5), 1-(3-dimethylaminopropyl)-3-ethyl-carbodiimide hydrochloride (EDC, ≥98%), *N*-hydroxysulfosuccinimide (NHS, 98%), allylamine (98%), toluene (anhydrous, 99.8%), formic acid (≥95%) and 1,4-dithioerythritol (≥99%) were purchased from Sigma-Aldrich (Zwijndrecht, The Netherlands). Poly(acrylic acid) (PAA, 25% soln. in water, *M_w_* ≈ 50 kg/mol) was purchased from Polysciences (Paris, France). *N*,*N*-Dimethylaminopropyl acrylamide (DMAPAA, 98%) was purchased from ABCR GmbH (Karlsruhe, Germany). Sodium metabisulfite (Na_2_S_2_O_5_, 98%) was purchased from Scharlau (Barcelona, Spain). The (3-Mercaptopropyl)trimethoxysilane (95%) was purchased from Alfa Aesar (Kandel, Germany). Methanol (99.9%), tetrahydrofuran (THF, stab./BHT, 99.8%), diethyl ether (stab./BHT AR, 99.5%) and acetonitrile (ACN, AR, 99.8) were purchased from Biosolve (Valkenswaard, The Netherlands). The 1.0 M and 0.1 M sodium hydroxide solutions (NaOH), 1.0 M and 0.1 M hydrochloric acid (HCl) solutions and CertiPUR^®^ (pH 4.0 buffer solution, citric acid/sodium hydroxide/hydrogen chloride) were purchased from Merck Millipore (Amsterdam Zuidoost, The Netherlands). Tetradecane (99%) was purchased from TCI Europe. Millipore water was obtained from Milli-Q (Millipore, conductivity: 0.055 mScm^−1^). Silicon wafers were purchased from ACM. Polyvinyl acetate glue (ref. L0196, 20 mL) was purchased from 3M. Cyanoacrylate adhesive (ref. 495) was purchased from Loctite (Boulogne-Billancourt, France). DMAPAA was passed through an alumina column to remove the inhibitor. All other products were used as received without further purification.

### 2.2. Polymer Synthesis

Poly(acrylic acid)-*g*-poly(*N*-isopropylacrylamide) (PAA-*g*-PNIPAM) (Figure 1A) was synthesized using a “grafting onto” technique according to the method developed by Durand [43]. Poly(*N*,*N*-dimethylaminopropyl acrylamide)*-g*-poly(*N*-isopropylacrylamide) (PDMAPAA-*g*-PNIPAM) (Figure 1B) was synthesized using a “grafting through” technique. First, a poly(*N*-isopropylacrylamide) macromonomer (macroPNIPAM) was synthesized according to the method developed by Petit [44] and subsequently copolymerized together with *N*,*N*-dimethylaminopropyl acrylamide (DMAPAA) to obtain the final copolymer. The detailed synthesis protocol can be found in our previous paper [25].

H-nuclear magnetic resonance spectroscopy (^1^H-NMR) measurements were performed in D_2_O on a Bruker Avance III 400 MHz NMR spectrometer in order to detect the molar ratio between PNIPAM and polyelectrolyte units. *M*_n_ of the anionic copolymer was also calculated via ^1^H-NMR, considering the initial molar mass of the PAA backbone, while *M_n_* of the cationic copolymer was determined by size exclusion chromatography (SEC) on an Agilent Technologies 1260 Infinity II system using a PSS Novema MAX 1000 Å column with an Agilent 1260 RI detector. Samples were run using water as eluent containing 300 mM formic acid at a flow rate of 0.6 mL min^−1^. The calibration was performed using poly(2-vinylpyridine) standards. In Table 1, the synthesized polymers are shown. The ^1^H-NMR spectra are shown in the Appendix A (Appendix A).

The apparent high polydispersity (PDI) of the cationic polymer is due to the free radical polymerization technique that does not allow precise control of the molecular weight distribution and due to the interactions of the polymer with the chromatography column, which lead to peak broadening in SEC, thereby increasing the apparent PDI.

### 2.3. Complex Coacervation

Stock solutions of PAA-*g*-PNIPAM and PDMAPAA-*g*-PNIPAM were prepared at a chargeable monomer concentration (PAA/PDMAPAA moles per unit volume) of 0.15 M. The pH of the PAA-*g*-PNIPAM solution was adjusted to 8.0 using 0.1 M NaOH and 0.1 M HCl. The LUDOX^®^ suspension was then added in the polyanion solution and the mixture was left to equilibrate in the refrigerator overnight. A volume of 5.0 M NaCl was added to the PDMAPAA-*g*-PNIPAM solution to adjust the ionic strength. Finally, a calculated amount of the PDMAPAA-*g*-PNIPAM solution, together with water, were added to the PAA-*g*-PNIPAM solution to reach in the final mixture a 0.05 M total chargeable monomer concentration, a 0.5 mixing ratio and a 0.75 M [NaCl]. The pH was then adjusted to 7.0 using 0.1 M NaOH and 0.1 M HCl. Complex coacervation took place directly after addition of the PAA-*g*-PNIPAM solution. After vigorous shaking, the complex coacervate phase was dispersed throughout the mixture. The mixture was left to equilibrate for 1 day and then it was centrifuged at 4000× *g* for 1 h. Two clearly separated phases appeared, with the complex coacervate phase sedimented at the bottom of the centrifuge tube. The complex coacervates were stored at 4 °C, well below the LCST.

### 2.4. Thermogravimetric Analysis (TGA)

The water and the nano-silica content in complex coacervates were investigated by thermogravimetric analysis (TGA) using a SDT Q600 from TA instruments. After removing the dilute phase from the Falcon^TM^ tube, the complex coacervate phase was directly loaded into the sample holder, a platinum pan, at room temperature. The samples were first equilibrated for 15 min at 110 °C. After that, they were subjected to a temperature ramp from 110 to 1200 °C at a heating rate equal to 20 °C/min.

### 2.5. Rheology

Rheological measurements were performed on an Anton Paar MCR301 stress-controlled rheometer using a cone-plate geometry (cone diameter 25 mm, cone angle 1°, measurement position 0.05 mm, glass plate). A Peltier element was used to regulate the temperature. The sample loading was performed as follows. The supernatant was taken off from the Falcon^TM^ tube using a Pasteur pipette, ending up with the complex coacervate phase only. We used this methodology because interested only in the mechanical properties of the complex coacervate phase when exposed to either a temperature or an ionic strength gradient. However, it is worth mentioning that upon removing the supernatant, the thermodynamical equilibrium is not affected: the complex coacervate phase remained stable and no free dilute phase was observed, i.e., the complex coacervate did not densify. This phase was then applied on the rheometer using a Pasteur pipette and contact with the cone was performed at the measurement position. When performing a temperature switch, tetradecane was added around the sample and a solvent trap with a metal lid was installed to prevent water evaporation. The temperature was then raised to 50 °C and a waiting time of 15 min was applied before any measurement. After removal of the dilute phase, the complex coacervate phase remained stable upon heating: the complex coacervate phase did not densify and no additional supernatant phase was expelled, so that two separate rheological measurements (below and above the LCST) were not necessary. When performing a salt switch, the lower ionic strength aqueous medium (0.1 M NaCl) was applied around the sample at 20 °C, with one-hour contact time before performing any rheological experiment. Before loading a new sample, the complex coacervate phase together with the dilute phase was centrifuged at 4000× *g* for 15 min.

#### 2.5.1. Linear Rheology

Amplitude sweeps were performed by varying the strain (*ε*) at a fixed angular frequency (*ω*, 1 rad/s) and at fixed temperature (20 °C) to determine the linear regime. It was observed that the storage (*G’*) and the loss (*G″*) moduli were constant almost over the whole range of amplitudes. A fixed strain equal to 1% was then set for all the following measurements.

Frequency sweeps were performed either at 20 °C or at 50 °C at a constant strain of 1% in a frequency range between 0.1 and 100 rad/s. Temperature sweeps were performed at a fixed frequency of 1 rad/s and at a fixed strain of 1% as the temperature was increased from 20 to 50 °C at a rate of 1 °C min^−1^.

#### 2.5.2. Non-Linear Rheology

Non-linear rheology was used to monitor the mechanical properties at high deformations above the LCST. The temperature was raised to 50 °C and an equilibration time of 60 min was applied. After that, shear start-up experiments were performed by shearing the samples at constant shear rate ($\dot{\gamma }$ = 0.1 s^−1^) and by monitoring the evolution of the shear stress (*σ*) as a function of strain (*ε*). Two replicas were conducted to ensure data reproducibility.

### 2.6. Differential Scanning Calorimetry (DSC)

The PNIPAM phase transition was investigated by DSC measurements using a Q200 from TA instruments. After removing the dilute phase from the FalconTM tube, the coacervate phase (~40 mg) was loaded into a Tzero Pan at room temperature. The samples, together with a reference filled with the same quantity of solvent (0.75 M NaCl solutions), were first equilibrated for 10 min at 15 °C. After that, they were submitted to temperature cycles from 15 to 60 °C, at a heating/cooling rate equal to 1 °C/min, which is the lowest scan rate above the sensitivity limit.

### 2.7. Underwater Adhesion

Underwater adhesion properties were measured using a tack test setup developed by Sudre et al. [45] and mounted on a Instron^®^ 5333 materials testing system with a 10 N load cell. The test consists of making a parallel contact and detachment underwater between a homogeneous layer of the complex coacervate (thickness ≈ 0.5 mm) and a poly(acrylic acid) (PAA) hydrogel thin film (thickness ≈ 200 nm).

The 5 × 5 mm wafer coated with the PAA hydrogel thin film was glued with a polyvinyl acetate adhesive to a mobile stainless-steel probe, which was fixed to the load cell and connected to the Instron machine. The complex coacervate sample was deposited onto a glass slide, which was previously fastened to the bottom of the chamber using plastic screws and aligned with the probe. Contact between the clean PAA thin film and the complex coacervate was performed at 20 °C until a 0.5 mm thickness was reached.

When performing a temperature switch, a 0.75 M NaCl water solution was poured in the chamber and the setup was covered at the top with a rubber layer providing heat insulation and temperature control. The whole chamber was heated to 50 °C using a temperature control equipment and the probe was kept motionless for 15 min. When performing a salt switch, a 0.1 M NaCl water solution was poured in the chamber at 20 °C and one-hour contact time between sample and probe was applied.

Detachment was then performed at a fixed strain rate of 0.2 s^−1^. Raw data of force and displacement were converted into stress and strain values to obtain the work of adhesion. The strain *ε* was obtained by normalizing the displacement by the initial thickness of the sample (*T*_0_). The normalized stress σ was obtained by dividing the force by the thin film contact area. The work of adhesion *W_adh_* was then calculated as follows:(1)$$W_{adh}=T_{0}\int_{0}^{\varepsilon_{max}}\sigma d\varepsilon$$

Three replicas were conducted for every experiment to ensure data reproducibility.

### 2.8. PAA Hydrogel Thin Film Synthesis

The PAA hydrogel thin film was synthesized by simultaneously crosslinking and grafting ene-functionalized poly(acrylic acid) (PAA) onto thiol-modified wafers through a thiol−ene click reaction according to the protocol developed by Chollet et al. [46]. The detailed protocol is reported in our previous work [25].

## 3. Results and Discussion

### 3.1. Complex Coacervation

The key challenge when developing a nano-reinforced material is to obtain a uniform dispersion of nanoparticles inside the matrix: due to their small size and high surface area, nanofillers interact strongly with each other through intermolecular forces, leading to aggregation [29]. This behavior prevents an effective reinforcement of the composite material, and thus the formation of such agglomerates has to be avoided. Silica nanoparticles are negatively charged when working at a pH above 8.0 and without any added salt: electrostatic repulsions therefore prevent aggregation. However, as explained in our previous work [25,26], a high sodium chloride (NaCl) concentration (0.75 M) and a pH of 7.0 are required to obtain a fluid complex coacervate phase that can be effectively used as an underwater adhesive: in these conditions the negative charges are almost completely screened and consequently bare silica particles aggregate through van der Waals interactions and sediment [36].

Crucially, this phenomenon is prevented by first adding the silica suspension to the anionic PAA-*g*-PNIPAM solution, prepared at pH 8.0 without any added salt: a hybrid network is then formed, with the pendant chains anchoring to the surface of the particles because of the strong adsorption of PNIPAM onto silica [36]. The physical interaction between PNIPAM and silica nanoparticles is very strong, leading to an immediate increase in the viscosity of the solution due to the formation of a physical network with silica beads acting as cross-linkers: this adsorption can be equilibrated in less than 24 h, thus, the solutions are stored overnight in the fridge to favour adsorption and reorganization of the chains on the surface [35]. Taking into account a maximum of 1mg/m^2^ for the adsorption isotherm of PNIPAM chains onto silica surfaces [35], all the premixed formulations of PAA-*g*-PNIPAM/LUDOX were prepared with an excess of PNIPAM in order to fully cover and saturate the silica surfaces with PNIPAM grafts. After one day, the oppositely charged polyelectrolyte solution is added and complex coacervation immediately occurs. Despite setting the NaCl concentration to 0.75 M and the pH to 7.0, only two phases (dilute phase + complex coacervate phase) can be observed, without any visible sign of macroscopic silica agglomeration (Figure 2, left). Conversely, if the silica dispersion is added after complex coacervate formation between the two oppositely charged polyelectrolyte solutions, clumps of silica, whose size increase as a function of the nanoparticle content, are formed, leading to undesirable sedimentation (Figure 2, right).

The presence of the PNIPAM chains, therefore, is crucial to allow the incorporation of the nanoparticles inside the complex coacervate matrix: by physically adsorbing onto the silica surfaces, they prevent macroscopic aggregation of the nanoparticles providing an effective steric protection, even in conditions of high ionic strength and intermediate pH [36,39]. Importantly, the adsorption equilibrium must be reached before complexation and a precise control of the mixing order and conditions are required to avoid formation of undesired nanoparticle agglomerates.

### 3.2. Thermogravimetric Analysis

The total SiO_2_ concentration in the mixture of the dilute phase and the complex coacervate phase has been set respectively to 0%, 0.1%, 0.5% and 1 w/w %. Interestingly, the partitioning of the silica nanoparticles in the two phases is not equivalent and upon complexation cannot be known a priori. Therefore, thermogravimetric analysis (TGA) is employed to define the final SiO_2_ concentration in the complex coacervate phase.

From Figure 3A three different steps are observed. The first huge decrease (0–200 °C) corresponds to the dehydration of the material, while the second step (200–600 °C) is the oxidation of the organic matter [47]. The last step (800–1000 °C) corresponds to the degradation of NaCl. When silica is added, the total mass does not reach zero even after 1000 °C; this is a further evidence that the nanoparticles are effectively retained in the complex coacervate phase and that the total content can be extracted from the TGA value obtained above 1000 °C. In Table 2 the characteristics of each analysed sample are reported, together with the name codes assigned (CCx, where CC stands for complex coacervate and x is the total SiO_2_ content in the complex coacervate).

Due to the presence of PNIPAM, the anionic copolymer chains can absorb onto the silica particle during the preparation stage. Since most of the polymer chains end up in the concentrated phase upon complexation, the silica nanoparticles partition preferentially in the complex coacervate phase: in every sample, more than 50% of the total silica ends up in the complex coacervate phase, with the concentration of nanoparticle, therefore, always being higher than in the dilute phase.

Additionally, from the TGA data the water content of the samples is also determined. In Figure 3B, it is observed that the water content decreases linearly with the silica concentration. The silica can be retained in the complex coacervate phase at the expense of water since both salt and polymer content, whose values can be obtained respectively by the second and third releasing step in the TGA, remain constant (Figure 3B, inset). The lower water retention might be explained by a higher tendency to phase separation, due to the PNIPAM adsorption onto the silica particles: when not bound to silica particles, the hydrophilic PNIPAM chains tend to contrast the tendency to phase separate due to complex coacervation, with a higher water content in PNIPAM-functionalized complex coacervates than the PNIPAM-free counterpart [25,26]. However, upon adding silica, since the fraction of the PNIPAM chains adsorbed onto the filler increase, the hydrophilic character decreases, therefore leading to lower water retention.

### 3.3. Rheology

#### 3.3.1. Linear Rheology

After formulating the hybrid complex coacervate adhesive, frequency sweeps are performed on the complex coacervates at 20 °C to determine the viscoelastic properties (Figure 4A).

By observing the frequency dependence of the moduli, it is possible to determine if the material has more liquid-like or solid-like properties. When no silica is present in the sample (CC0), the complex coacervate shows features of a viscous liquid: the storage modulus (*G’*) is lower than the loss modulus (*G″*) up to high frequencies, where a crossover is detected. The chains can slide along each other with transient electrostatic interactions [48] and with a short terminal relaxation time *τ*, which can be obtained as the inverse of the crossover frequency (*ω_c_*). By adding silica nanoparticles to the materials, the moduli rise and become increasingly less frequency dependent. Additionally, the crossover shifts gradually to lower frequencies, corresponding to an increase in relaxation time. Up to a certain threshold, CC3.5, the crossover is no longer visible with *G’* exceeding *G″* over the entire measured range of frequencies. These data suggest that the addition of silica nanoparticles greatly slows down the chain dynamics with the material behaving like a soft gel at a high silica content. The tremendous increase in dynamic moduli is due to the adsorption of PNIPAM chains onto the silica beads leading to the formation of hybrid gels, as already reported in literature: a network can be created by forming new junctions between the polymeric chains and the nanoparticles [35,36,49]. The sol-gel transition can be obtained by raising the silica concentration and can be observed when the number of connections between polymer chains exceeds the percolation threshold: in this case, the critical value is approximately 3.5% w/w in silica nanoparticles, in good accordance with literature data [35].

While at room temperature the viscoelastic behavior is mainly dominated by hybrid crosslinks, above the LCST, both PNIPAM–silica and PNIPAM–PNIPAM interactions contribute to the rheological properties [50]. This can be clearly observed in the temperature sweeps reported in Figure 4B. The dynamic moduli of all hybrid complex coacervates are enhanced at higher temperature [35], meaning that, in every case, free PNIPAM chains can undergo the phase transition. However, the crossover between *G’* and *G″* can only be observed at low silica concentration, while higher silica concentrations exist above the percolation threshold and the material behaves as a solid gel already at room temperature. Additionally, the onset of the thermal transition gradually shifts to higher temperatures when increasing the nanofiller content: at a higher concentration of silica, the fraction of PNIPAM chains adsorbed onto the surface is higher so that a higher energy and, therefore, a higher temperature is required to induce the dehydration process of the remaining PNIPAM units [35,36]. This is also confirmed by the DSC data, which report a gradual LCST increase from 25.7 °C (CC0) to 26.8 °C (CC6). The phase transition is completely reversible: upon cooling, the dynamic moduli get back to the original values, with a heating rate-dependent hysteresis effect due to the different kinetics of the association and dissociation processes of the thermoresponsive chains. In accordance with our previous results [25], neither the water content nor the volume of the complex coacervate phase change during the whole process.

At 50 °C, as evidenced in Figure 5A, the final moduli have nearly the same values and the same frequency dependence, no matter the silica content. At a higher silica concentration, there are fewer PNIPAM units able to undergo the phase transition above the LCST. However, since there is always an excess of free PNIPAM chains that are not involved in the adsorption process, at high temperature the viscoelastic properties are mainly dominated by the associations between the thermoresponsive units [36,50].

The sol-gel transition can also be observed when decreasing the ionic strength of the surrounding environment at a constant temperature: Figure 5B shows the frequency sweeps performed on the hybrid complex coacervate phase after a 1 h equilibration time at room temperature with a 0.1 M NaCl aqueous solution in the rheometer. The material turns into a soft gel (both *G’* and *G″* are almost frequency independent) as the formation of stronger ionic bonds between oppositely charged polyelectrolytes at lower ionic strength: the salt present in the complex coacervate phase, responsible for the screening of most of the electrostatic interactions between the backbones, diffuses out of the adhesive in response to a gradient in ionic strength [26,51]. The addition of the nanofillers in the complex coacervate phase has the effect of increasing the dynamic moduli. This behaviour can be explained using an argument from the statistical theory of rubber elasticity [52], according to which the shear modulus of a network is proportional to the number of elastic strands, and therefore crosslinks, per unit volume; this explanation is also consistent with the observed temperature switch. Two kinds of physical crosslinking units are present in this case. The first are the ionic bonds between oppositely charged polyelectrolytes: since the polymer concentration in the material is the same (as detected by TGA), the number of interpolyelectrolyte interactions does not vary as a function of the silica content. However, by adding silica nanoparticles, PNIPAM chains adsorb onto the nanofiller, creating new physical crosslinks: as a consequence, the higher the silica content the greater the number of interactions, leading to a progressive increase in the moduli. We can also postulate the presence of some electrostatic interactions between the cationic polymer and the negatively charged silica particles: the amount of these interactions might, however, be negligible if we assume that the nanoparticles are fully covered with PNIPAM side chains with very weak dynamics. Differently from the temperature switch dominated by hydrophobic PNIPAM/PNIPAM interactions, the contribution of hybrid PNIPAM/silica junctions on the viscoelastic properties is more visible after the salt switch as the number of electrostatic cross-links created at low ionic strength is much lower than previous hydrophobic ones.

The type of switch performed also affects the extent of dissipation in the material, which can be quantified by normalizing, at a fixed angular frequency, the *tan δ* (*G″*/*G’*) by the storage modulus recorded at the fixed frequency (Figure 5C) [53]. It is observed that the amount of dissipation is higher, at any silica content, when performing a salt switch: this is expected to considerably affect the mechanical behavior in non-linear deformations (e.g., adhesion test), as we will highlight in the following sections.

#### 3.3.2. Non-Linear Rheology

The toughness of these hybrid complex coacervates is determined using a shear start-up experiment performed on complex coacervates after raising the temperature above the LCST (Figure 6).

From the stress–strain curves, the low strain region, corresponding to a linear deformation, as shown in Figure 6 inset, is in agreement with the values of *G’* found during the frequency sweeps at 50 °C as seen in Figure 5A: the slope of the curve, which represents the shear modulus, slightly increases with added nanofiller and remains constant at higher silica content. However, both the stress and the strain at break decrease when the silica content is higher. Remarkably, the addition of the silica nanoparticles has an unexpected negative effect on the non-linear mechanical properties, decreasing both fracture resistance and toughness.

This behavior has been observed in hybrid hydrogels in which the nanoparticles interact weakly with the matrix [54]: the filler in this case can be embedded in the network only up to a certain threshold (7% volume fraction), after which the system phase separates because of the aggregation of the nanoparticles. Below that critical concentration, the fracture resistance decreases as a function of the filler content, as observed in our work. By approaching the phase separation boundary, the system becomes gradually more heterogeneous, with regions of high and low concentration of nanoparticles: at the interface then, fracture planes may form, which would promote earlier failure of the network. However, in this work, the polymeric chains interact much more strongly with the filler and, at a higher silica content, the PNIPAM coverage of the nanoparticles gradually decreases [35,36]. As a consequence, the steric stabilization decreases, leading to a more heterogeneous distribution of nanoparticles in the complex coacervate phase, as observed in the case of weak interactions between polymeric matrix and fillers [54]: fracture planes easily form, leading to lower toughness at a high silica content.

The heterogeneity might also originate from the mixing process: the strong affinity between silica nanoparticles and PAA-*g*-PNIPAM leads to a very fast interaction as soon as the two components are mixed. As a consequence, it is hard to obtain a homogeneous formulation, as already reported by Petit et al. [35], which, for this reason, did not explore polymer concentrations above 2% wt.

Another consequence of the strong interaction between silica and PNIPAM is that the PAA-*g*-PNIPAM chains are tightly wrapped around the silica particles. This condition may lead to a situation where negatively charged silica/PAA-*g*-PNIPAM spheres are incorporated in a positively charged PDMAPAA-*g*-PNIPAM matrix. In this scenario, the only interactions that need be broken upon material deformation are the electrostatic interactions between charged domains negating any effective contribution from the strong PNIPAM–silica junctions. Due to their confinement close to the silica surface, PAA-*g*-PNIPAM chains cannot move freely, preventing optimal complexation with PDMAPAA-*g*-PNIPAM chains, as opposed to the silica free scenario, in which the oppositely charged polymer chains can make bonds with several electrostatic interactions in a row. Therefore, upon adding more silica, less electrostatic interactions are formed, apparently resulting in a decrease in toughness as seen in Figure 6.

### 3.4. Underwater Adhesion

Underwater adhesion experiments are performed on the complex coacervates using a tensile probe-tack geometry after raising the temperature above the LCST (Figure 7).

The adhesive performance decreases as a function of the silica content: the stress peak drops progressively to lower values at a much lower strain (Figure 7A), causing a decrease in the work of adhesion (Figure 7B). The observed mode of failure is always cohesive, with residues of samples on both the probe and adherent. Interestingly, the presence of an additional energy-dissipating mechanism due to the additional nanofiller does not improve the final performance. As evidenced in the rheology section, the mechanical properties are mainly dominated by the PNIPAM–PNIPAM interactions, so that a gain in toughness at a higher filler content is not expected. Additionally, the material becomes increasingly more heterogeneous and failure occurs more easily at the interface between regions of high and low concentration of nanoparticles. Lastly, when increasing the silica content, the complex coacervate becomes more elastic, as seen in Figure 5C, compromising its ability to form fibrils and consequently to dissipate energy: it becomes easier to form cracks and grow them perpendicular to the tensile direction.

By contrast, when applying a trigger in salt concentration instead of temperature, the scenario is dramatically different (Figure 8).

At low silica concentrations, a better performance (Figure 8A) and an increase in the work of adhesion (Figure 8B) can be observed up to a certain threshold (3.5 w/w %): this behavior can be ascribed to the higher cohesive properties in presence of PNIPAM–silica junctions, which provide additional strength to the material, as evidenced already in the linear rheology measurements. Additionally, when silica is present in the material, the curves have an atypical shape: the stress rapidly increases and reaches a peak at very low strain, after which a plateau is observed at lower stress values up to very high deformations. At the end of the test, the stress drops to zero because of the failure of the material. The first peak might be related to the failure of the PNIPAM–silica bonds, as this is not visible in the silica-free complex coacervates. The plateau observed at high strains instead can be ascribed to the stretching of the polyelectrolyte backbones, which are tightly connected through electrostatic interactions and can only be broken when reaching high deformations. Despite an increasing heterogeneity of formulations, the addition of silica particles improves the toughness and, therefore, underwater adhesion performance through an additional energy dissipation mechanism.

Additionally, the hybrid material always fails cohesively and can be stretched to much higher strains compared to a temperature trigger. As already reported in our previous work [26], this is probably due to the architecture of the polymer chains. When using a temperature trigger, interactions between short PNIPAM chains are activated: the formed domains are then expected to be small and cannot be stretched to a very high extent. However, when using a salt trigger, ionic interactions between long polyelectrolyte chains are formed: the larger domains formed can sustain stress at very high deformations. Conversely, when the silica concentration reaches too high values (6%), the gain in strength is not sufficient to contrast the formation of fracture planes due to the increased heterogeneity of the material, leading to premature failure. Nevertheless, the work of adhesion, at any silica content, is always higher when performing a salt switch compared to the temperature switch: this might be related to the higher *tan δ/G’* ratio observed when using a salt trigger (Figure 5C), resulting in a higher energy dissipation and consequently a better adhesion performance.

## 4. Conclusions

In this work, we have successfully incorporated silica nanoparticles into complex coacervates without visible aggregation or phase separation of these nanofillers. Below the LCST, the dynamic moduli of these hybrid materials increase as a function of the silica content due to the adsorption of PNIPAM chains onto the nanoparticles. However, above the LCST, the PNIPAM–silica junctions do not make a significant contribution to the enhancement of the mechanical properties, which are mainly dominated by the association of free PNIPAM chains. A larger work of adhesion is instead observed when performing a salt switch, with the moduli increasing as a function of the amount of PNIPAM–silica junctions, which contribute to the reinforcement of the material together with the stronger electrostatic interactions between the polyelectrolyte backbones leading to, at best, a two-fold improvement of the work of adhesion.

However, the silica content needs to be carefully tuned to prevent both a non-uniform dispersion in the polymeric matrix and an excessive reinforcement of the moduli already before the adhesive application, dramatically undermining the final performance which occurs in hybrid materials above a threshold silica content of ~4%. Surprisingly, the addition of silica nanoparticles into a complex coacervate matrix is not an efficient method toward a high-performance underwater adhesive given the complexity of the system and the only modest improvement of the mechanical properties.

These results might be strongly affected by the type of filler used in this work: the incorporation of a neutral filler, which does not interact with the charged backbones and whose stability is not affected by salt concentration and pH, might result in an improved performance. This would pave the way for the development of nano-reinforced complex coacervates which can be effectively used as injectable adhesives for biomedical purposes, such as wound closure or soft tissue repair.

## 5. Patents

A patent application about part of the work reported in this paper has been filed and is under revision.

## Figures and Tables

**Figure 1 polymers-12-00320-f001:**
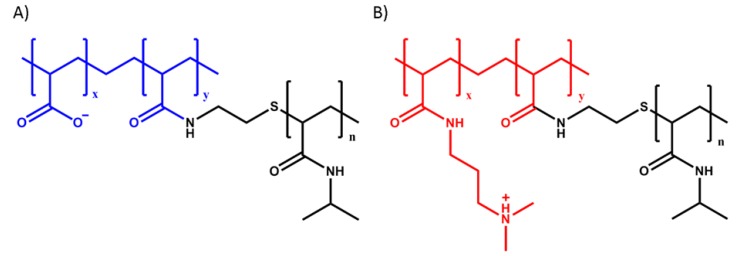
Molecular structure of (**A**) PAA-*g*-PNIPAM and (**B**) PDMAPAA-*g*-PNIPAM. The colored parts represent the polyelectrolyte backbones, while the black ones represent the PNIPAM units.

**Figure 2 polymers-12-00320-f002:**
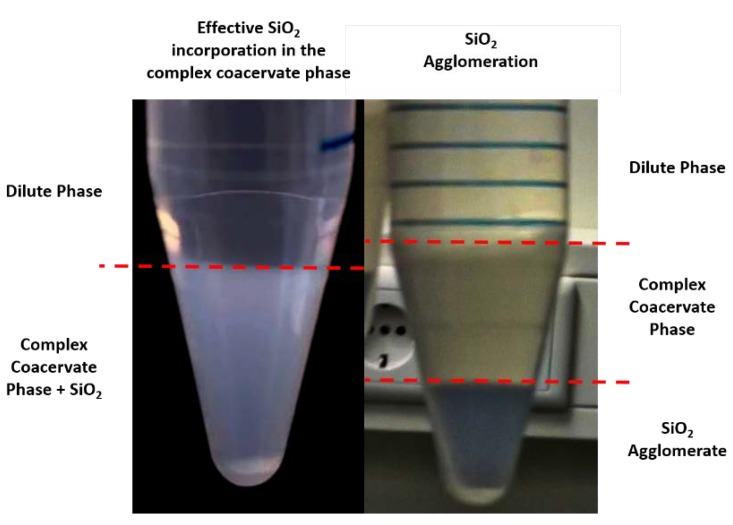
**Left**: effective incorporation of the silica nanoparticles inside the complex coacervate phase. **Right**: silica aggregation when the suspension is added after complex coacervation.

**Figure 3 polymers-12-00320-f003:**
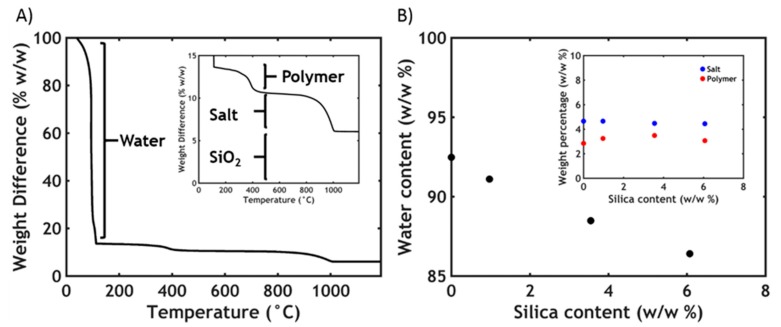
(**A**) Typical thermogravimetric analysis (TGA) thermogram (in the inset, zoom in the area between 0% and 15% w/w) and (**B**) water content plotted as a function of silica concentration in silica-containing complex coacervates (in the inset, salt and polymer content plotted as a function of silica content).

**Figure 4 polymers-12-00320-f004:**
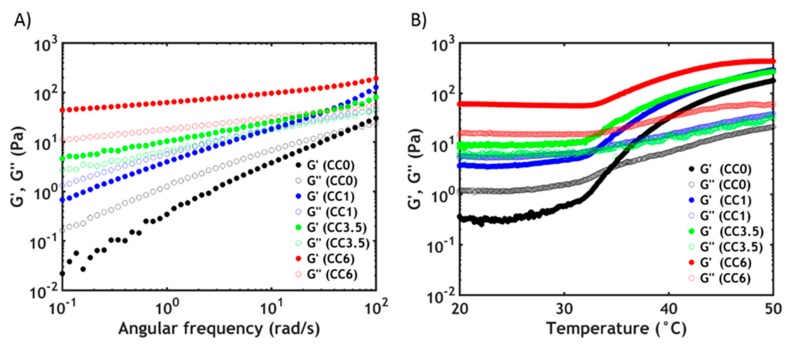
(**A**) Frequency sweeps performed on complex coacervates at 20 °C. (**B**) Temperature sweeps at *ω* = 1 rad/s. The full dots represent *G’*, the hollow dots represent *G″*.

**Figure 5 polymers-12-00320-f005:**
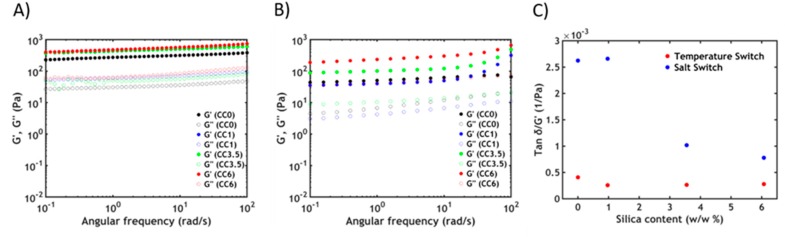
Frequency sweeps performed on complex coacervates after (**A**) a temperature triggered setting process at 50 °C and (**B**) after a salt-triggered setting process by reducing [NaCl] = 0.1 M. The full dots represent *G’*, the hollow dots represent *G″*. (**C**) The ratio *tan δ/G’* recorded at 1 rad/s represented as a function of silica content for both switches.

**Figure 6 polymers-12-00320-f006:**
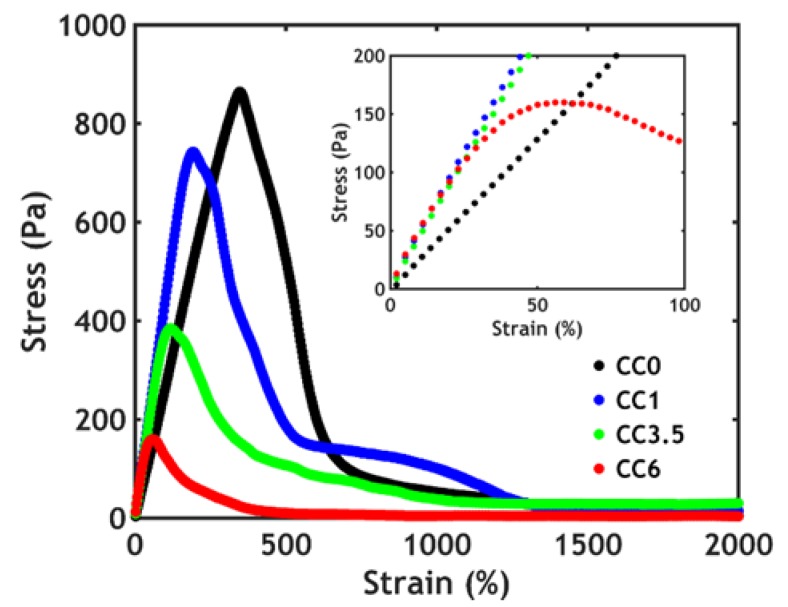
Shear start-up experiments performed on complex coacervates at 50 °C.

**Figure 7 polymers-12-00320-f007:**
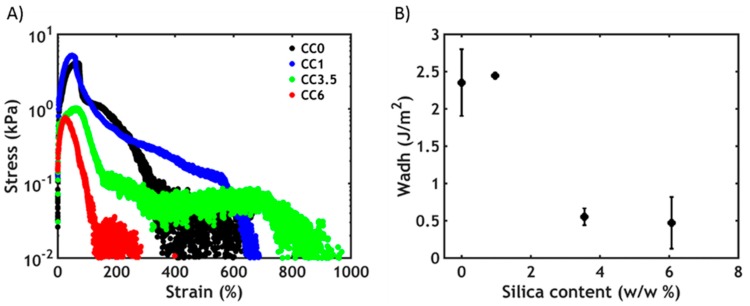
Underwater adhesion experiments after a temperature-activated setting process: (**A**) stress–strain curves and (**B**) work of adhesion as a function of silica content.

**Figure 8 polymers-12-00320-f008:**
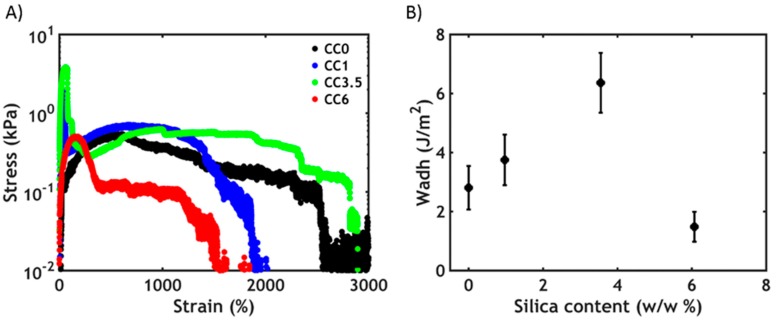
Underwater adhesion experiments after a salt-activated setting process: (**A**) stress–strain curves and (**B**) work of adhesion as a function of silica content.

**Table 1 polymers-12-00320-t001:** Graft copolymers synthesized in this work.

Polymer	PNIPAM/Total Polymer Molar Ratio (%)	*M_n_* Graft Copolymer (kg/mol)	PNIPAM Chains per Backbone	PDI
PAA-*g*-PNIPAM	42	588	51	-
PDMAPAA-*g*-PNIPAM	26	248	7	4.41

**Table 2 polymers-12-00320-t002:** Complex coacervates analyzed in this study.

Sample Name	[SiO_2_] in Mixture (w/w %)	[SiO_2_] in Complex Coacervate Phase (w/w %)	[SiO_2_] in Dilute Phase (w/w %)	Percentage of SiO_2_ Ending in Complex Coacervate Phase (w/w %)
CC0	0	0	0	0
CC1	0.1	0.97	0.01	87
CC3.5	0.5	3.55	0.2	64
CC6	1	6.07	0.49	55

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
