# Peer review of "Hybrid Complex Coacervate"

_polymers, 2020, doi:10.3390/polym12020320_

Round 1
Reviewer 1 Report
Dompe et al. prepared the hybrid complex coacervate with SiO2 and investigated the rheological properties including linear and non-linear, and underwater adhesive property. They successfully showed the SiO2-incorporated complex coacervate phase using the PNiPAM moiety on the polymer matrix, where the PNiPAM moiety acts as steric stabilizer by adsorbing on the SiO2 surface. Then they measure rheological properties and underwater adhesive properties of the hybrid complex coacervate layer after removing supernatant. Although the preparation and their behavior are highly interesting, I do have some substantial concerns as shown below:
(1) As shown in the experimental section, when they measure the rheological properties of the layer upon heating, they removed the supernatant phase intentionally. I think the hybrid complex coacervate always wants to achieve themodynamically equilibrium state when circumstances change. That is, when the supernatant is removed, then new equilibrium is reached, new supernatant is formed and the complex coacervates become denser. Authors would like to make a comment on this issue.
(2) In prior to temperature dependent measurement, authors may want to measure the reversibility of the phase upon temperature. It is unclear if the volume of complex coacervate + SiO2 layer changes upon changing temperature around LCST of PNiPAM.
(3) After removal of supernatant, I am wondering if the phase separation occurs above LCST of PNiPAM during the rheological measurement. If the phase separation occurs above LCST, they should measure separately below LCST and above LCST.
(4) The resolution of Figure 2 should be improved.
(5) Line 292-294: author may want to explain why the water content decreases as SiO2 increases. Is there any role of SiO2 which shows similarity to water?
Reviewer 2 Report
The ms is suitable for publication before some minor concerns. In particular, I believe the authors need to put more emphasis on Conclusions section on the possibile future applications of the results.
Round 2
Reviewer 1 Report
Authors have provided reasonable answers to all the questions and modifications on the manuscript, and thus I recommend the publication of the manuscript in Polymers.